# Protein Thermal Stability Changes Induced by the Global Methylation Inhibitor 3-Deazaneplanocin A (DZNep)

**DOI:** 10.3390/biom14070817

**Published:** 2024-07-09

**Authors:** Christine A. Berryhill, Emma H. Doud, Jocelyne N. Hanquier, Whitney R. Smith-Kinnaman, Devon L. McCourry, Amber L. Mosley, Evan M. Cornett

**Affiliations:** 1Department of Biochemistry and Molecular Biology, Indiana University School of Medicine, Indianapolis, IN 46202, USAedoud@iu.edu (E.H.D.); johanqui@iu.edu (J.N.H.);; 2Center for Proteome Analysis, Indiana University School of Medicine, Indianapolis, IN 46202, USA; 3Indiana University Simon Comprehensive Cancer Center, Indiana University School of Medicine, Indianapolis, IN 46202, USA; 4Center for Computational Biology and Bioinformatics, Indiana University School of Medicine, Indianapolis, IN 46202, USA

**Keywords:** lysine methylation, S-adenosyl-l-methionine, methionine cycle, proteome integral stability alteration assay

## Abstract

DZNep (3-deazaneplanocin A) is commonly used to reduce lysine methylation. DZNep inhibits S-adenosyl-l-homocysteine hydrolase (AHCY), preventing the conversion of S-adenosyl-l-homocysteine (SAH) into L-homocysteine. As a result, the SAM-to-SAH ratio decreases, an indicator of the methylation potential within a cell. Many studies have characterized the impact of DZNep on histone lysine methylation or in specific cell or disease contexts, but there has yet to be a study looking at the potential downstream impact of DZNep treatment on proteins other than histones. Recently, protein thermal stability has provided a new dimension for studying the mechanism of action of small-molecule inhibitors. In addition to ligand binding, post-translational modifications and protein–protein interactions impact thermal stability. Here, we sought to characterize the protein thermal stability changes induced by DZNep treatment in HEK293T cells using the Protein Integral Solubility Alteration (PISA) assay. DZNep treatment altered the thermal stability of 135 proteins, with over half previously reported to be methylated at lysine residues. In addition to thermal stability, we identify changes in transcript and protein abundance after DZNep treatment to distinguish between direct and indirect impacts on thermal stability. Nearly one-third of the proteins with altered thermal stability had no changes at the transcript or protein level. Of these thermally altered proteins, CDK6 had a stabilized methylated peptide, while its unmethylated counterpart was unaltered. Multiple methyltransferases were among the proteins with thermal stability alteration, including DNMT1, potentially due to changes in the SAM/SAH levels. This study systematically evaluates DZNep’s impact on the transcriptome, the proteome, and the thermal stability of proteins.

## 1. Introduction

Lysine methylation (Kme) is a reversible post-translational modification (PTM) detected on upwards of 5000 human proteins [1]. Lysine methylation regulates protein–protein interactions, protein–DNA interactions, protein stability, and protein activity [2]. The enzymes that add lysine methylation, lysine methyltransferases (KMTs), are frequently dysregulated in many cancers, making them attractive therapeutic targets [3,4]. Therefore, several inhibitors have been created or adapted to modulate the activity of these enzymes. KMTs catalyze the transfer of up to three methyl groups from the universal methyl donor S-adenosly-l-methionine (SAM) to a target lysine within a protein, producing S-adenosylhomocysteine (SAH) as a byproduct. Many inhibitors directly bind to the substrate binding site or SAM binding site of specific KMTs [3]. An indirect way to modulate the activity of KMTs is by decreasing the amount of SAM via the inhibition of critical components of the methionine cycle [5,6]. One such inhibitor that is widely used as a research tool is 3-deazaneplanocin A (DZNep), which inhibits Adenosylhomocysteine hydrolase (AHCY), a critical enzyme of the methionine cycle that converts SAH to homocysteine and adenosine using nicotinamide adenine dinucleotide (NAD+) as a cofactor [7,8,9]. DZNep is a potent inhibitor of AHCY in vitro with a reported Ki of 0.05 nM [9]. Because it is the only enzyme that converts SAH to homocysteine, inhibiting AHCY increases the concentration of SAH and decreases the SAM-to-SAH ratio, an indicator of the methylation potential within a cell [9].

In multiple studies, DZNep has been evaluated as a potential therapeutic option for many cancer types through the modulation of the aberrant histone methylation common in cancer [10,11,12]. These studies discovered that DZNep depletes endogenous levels of EZH2, a lysine methyltransferase that deposits di- and trimethylation on histone 3 lysine 27 (H3K27me2/3) [12]. The removal of these repressive marks induces the expression of apoptosis-related genes, leading to the apoptosis of treated cells [12,13]. Given the depletion of EZH2 caused by DZNep, it is frequently cited as an EZH2 inhibitor, though it also decreases H3K9me2, H4K20me3, and H3K36me3 levels [14]. Given that the enzymes that mediate histone methylation frequently have non-histone substrates, DZNep likely also impacts non-histone proteins that are regulated by lysine methylation. Additionally, because DZNep is an adenosine analog, there may be substantial off-target effects due to binding to any protein with an affinity for AMP/ADP/ATP, SAM, or other nucleotide analogs. To address this, one study created a functional derivative of DZNep and performed a pull-down that identified 41 interacting proteins, including kinases, phosphatases, ATPases, and methyltransferases, suggesting DZNep interacts with proteins other than AHCY [15]. However, no studies have investigated the downstream impact beyond the direct targets of DZNep.

In this study, we investigated the impact of DZNep on the transcriptome, proteome, and thermal stability of proteins. In recent years, protein thermal stability assays have been used to identify and study how post-translational modifications, including phosphorylation and O-linked N-acetylglucosamine (O-GlcNAc), impact the thermal stability of specific modified proteoforms [16,17]. The basic principle underlying the thermal stability assay is that a given proteoform aggregates at a certain temperature [18]. Collecting the soluble proteins along a temperature gradient and using liquid chromatography–tandem mass spectrometry (LC-MS/MS) enables the generation of a melting curve for thousands of proteins simultaneously. A recent adaptation, called the Proteome Integral Solubility Alteration (PISA) assay, involves pooling all the soluble protein into a single sample, allowing for the analysis of the integral of the melt curve [19]. This enables multiple replicates to be run simultaneously, increasing the number of proteins detected between the replicates and decreasing the inherent variability between runs. In this study, we used PISA to investigate the impact of DZNep on the thermal stability of methylated proteins. Our results show that DZNep impacted the thermal stability of over 100 proteins while inducing large changes to the transcriptome and proteome.

## 2. Materials and Methods

### 2.1. Cell Culture and Drug Treatment

The HEK293T cells were purchased from the ATCC (Manassas, VA, USA). They were cultured in RPMI media with 10% fetal bovine serum and 1× Penicillin/Strep at 37 °C with 5% CO_2_. For all the experiments, two million cells were seeded into a 10 cm dish 24 h prior to treatment with DZNep (5, 10, 25, 50, and 100 μM). The cells were collected 48 h later and immediately subjected to the Proteome Integral Solubility assay.

### 2.2. Western Blots

The cells were grown until 90% confluent and pelleted, and the protein was extracted by resuspending it in lysis buffer (10 mM Pipes pH 7, 300 mM sucrose, 100 mM NaCl, 3 mM MgCl_2_, 0.1% Triton X-100, protease inhibitors (Thermo Scientific, Waltham, MA, USA—Pierce #A32965), and nuclease (Thermo Scientific—Pierce #88700)) and incubating for 15 min on ice. The samples were quantified using a Bradford Assay (BioRad, Hercules, CA, USA). The protein lysates (30 μg) were separated by 8% SDS-PAGE and transferred to a polyvinylidene difluoride (PVDF) membrane using a semi-dry transfer. The membranes were blocked for one hour at room temperature in 5% non-fat dry milk in PBS with 0.1% Tween20 and probed antibodies overnight at 4 °C with anti-pan monomethyl lysine (Cell Signaling Technology, Danvers, MA, USA #14679), anti-pan dimethyl lysine (Cell Signaling Technology #14117), anti-pan trimethyllysine (Cell Signaling Technology #14680), or tubulin (Proteintech, Rosemont, IL, USA 66240-1-Ig).

### 2.3. Proteome Integral Solubility Assay

The HEK293T cells were treated with either 100 μM DZNep or vehicle control for 48 h (*n* = 4). The cells were harvested in 1 mL of ice-cold PBS. A total of 1 million cells were split off and collected via flash freezing for RNA sequencing. An equal number of cells were resuspended in 500 μL of PBS with the Pierce Protease Inhibitor and aliquoted equally into ten PCR tubes (50 μL each). A total of 8 tubes were heat-treated for 3 min along a gradient (39.4, 43.8, 47.1, 51. 53.4, 55.4, 57.9, and 61.4 °C) followed by 25 °C for 2 min in a BioRad C1000 Touch Thermal Cycler. The heat-treated and one non-heat-treated tube (global sample) were lysed via 3× freeze–thaw cycles. The samples were transferred into 1.5 mL microtubes and spun down at 17,000× *g* for 10 min, and the supernatant was transferred into another 1.5 mL microtube. A total of 20 μL from each heat-treated sample was combined into a new tube (160 μL total) and frozen until processing for the LC-MS/MS analysis as described below.

### 2.4. RNAseq

The samples were submitted to Azenta for extraction, library preparation, and sequencing. The concentration and quality of each RNA sample was assessed using RNA ScreenTape. The extracted RNA was sequenced on the Illumina (San Diego, CA, USA) HiSeq with paired reads for 150 bp. The sequences were trimmed using Trimmomatic v.0.36. The trimmed reads were then mapped to the homo sapiens GRCh38 reference genome using the STAR aligner v.2.5.2b. The unique gene counts were calculated using featureCounts from the Subread package v.1.5.2. Only the unique reads that fell within exon regions were counted. A differential gene expression analysis was performed using DESeq2. The Wald test was used to generate *p*-values and log_2_ fold changes (FC), and significantly differential genes were identified with the cutoff of an adjusted *p*-value < 0.05 and an absolute log_2_ fold change > 1.

### 2.5. Proteomics Sample Preparation and Nano-LC-MS/MS

The sample preparation, mass spectrometry analysis, bioinformatics, and data evaluation for the quantitative proteomics and phosphoproteomics experiments were performed in collaboration with the Indiana University Proteomics Center for Proteome Analysis at the Indiana University School of Medicine similarly to previously published protocols [20,21]. The cells collected for the global and PISA analysis, as described above, were quantified using a Bradford Assay (BioRad). A total of 30 µg equivalent of protein from each sample was precipitated using 20% final volume trichloroacetic acid (overnight at 4 °C followed by centrifugation at 12 k rcf for 30 min and two washes with acetone). The protein pellets were then resuspended in 30 µL 8 M Urea and 100 mM Tris pH 8.5 treated with 5 mM tris(2-carboxyethyl)phosphine hydrochloride (Sigma-Aldrich, St. Louis, MO, USA, Cat No: C4706) to reduce the disulfide bonds, and the resulting free cysteine thiols were alkylated with 10 mM chloroacetamide (Sigma Aldrich Cat No: C0267). The samples were diluted with 50 mM Tris HCl pH 8.5 (Sigma-Aldrich Cat No: 10812846001) to a final urea concentration of 2 M for overnight trypsin/Lys-C digestion at 35 °C (1:50 protease:substrate ratio, mass spectrometry grade, Promega Corporation, Madison, WI, USA Cat No: V5072). The digestions were acidified with trifluoroacetic acid (TFA, 0.5% *v*/*v*) and desalted on Waters Sep-Pak^®^ Vac cartridges (Waters™, Milford, MA, USA Cat No: WAT054955) with a wash of 1 mL 0.1% TFA followed by elution in 0.6 mL of 70% acetonitrile 0.1% formic acid (FA). The peptides were dried by speed vacuum and resuspended 50 mM triethylammonium bicarbonate. Each sample was then labeled for two hours at room temperature, with 0.5 mg of Tandem Mass Tag Pro (TMTpro) reagent (16-plex kit, manufactures instructions Thermo Fisher Scientific, TMTpro™ Isobaric Label Reagent Set; Cat No: 44520, lot no. VF300895) [22]. The reactions were quenched with 0.3% hydroxylamine (*v*/*v*) at room temperature for 15 min. The labeled peptides were then mixed and dried by speed vacuum. Half of the combined sample was resuspended in 0.5% TFA and fractionated on a Waters Sep-Pak^®^ Vac cartridge (Waters™ Cat No: WAT054955) with a 1 mL wash of water, 1 mL wash of 5% acetonitrile, and 0.1% triethylamine (TEA), followed by elution in 12.5%, 15%, 17.5%, 20%, 22.5%, 25%, 30%, and 70% acetonitrile, all with 0.1% TEA. Mass spectrometry was performed utilizing an EASY-nLC 1200 HPLC system (SCR: 014993, Thermo Fisher Scientific) coupled to an Exploris 480™ mass spectrometer with the FAIMSpro interface (Thermo Fisher Scientific). The technical replicates were run, first with 1/5th of each fraction and then with 1/3rd of each fraction onto a 25 cm EasySpray column (Thermo Fisher ES902) at 350 nL/min. The gradient (mobile phases A: 0.1% formic acid (FA), water; B: 0.1% FA, 80% Acetonitrile (Thermo Fisher Scientific Cat No: LS122500)) was increased from 8 to 38% B over 98 min and 30 to 80% B over 10 min; held at 80% for 2 min; and dropped from 80 to 4% B over the final 5 min. The mass spectrometer was operated in the positive ion mode, with a default charge state of 2, the advanced peak determination on, and a lock mass of 445.12003. Three high-field asymmetric waveform ion mobility spectrometry (FAIMS) compensation voltages (CVs) were utilized (−40 CV; −55 CV; and −70 CV) each with a cycle time of 1.3 s and with identical MS and MS2 parameters. The precursor scans (*m*/*z* 375–1500) were performed with an Orbitrap resolution of 120,000, radio frequency lens% 40, an automatic maximum inject time, a standard automatic gain control target, a minimum MS2 intensity threshold of 5 × 10^3^, and the monoisotopic precursor selection mode to the peptide, including charges of 2 to 7 for fragmentation with 30 s dynamic exclusion. The MS2 scans were performed with a quadrupole isolation window of 1.6 *m*/*z*, 30% higher energy collision dissociation normalized collision energy, a 15,000 resolution, a standard automatic gain control target, an automatic maximum injection time, and a fixed first mass of 110 *m*/*z*.

### 2.6. Proteomic Data Analysis

The resulting RAW files were analyzed in Proteome Discover™ 2.5 (Thermo Fisher Scientific) with a Homo sapiens UniProt reviewed proteome FASTA (downloaded 051322, 20,282 sequences) plus common laboratory contaminants [23]. SEQUEST HT searches were conducted with a full trypsin digest, including 4 maximum number missed cleavages; a precursor mass tolerance of 10 ppm; and a fragment mass tolerance of 0.02 Da. The static modifications used for the search were (1) carbamidomethylation on cysteine (C) residues and (2) a TMTpro label on N-termini of peptides. The dynamic modifications used for the search were the TMTpro label on lysine (K) residues, methylation on lysine (K) residues, demethylation on lysine (K) residues and trimethylation on lysine (K) residues, oxidation of methionines, and methionine loss or acetylation with methionine loss on protein N-termini. The Percolator False Discovery Rate (FDR) was set to a strict setting of 0.01 and a relaxed setting of 0.05. An IMP-ptm-RS node was used for all the modification site localization scores. The values from both the unique and razor peptides were used for quantification. In the consensus workflows, the peptides were normalized by the total peptide amount with no scaling. The quantification methods utilized the TMTpro isotopic impurity levels available from Thermo Fisher Scientific. Reporter ion quantification was allowed with a signal-to-noise threshold of 6 and co-isolation threshold of 30%. The resulting grouped abundance values for each sample type from Proteome Discover were exported and analyzed as described below.

### 2.7. Data Analysis

All of the data analysis was conducted in R v.4.3 using in-house scripts. The proteins were filtered for an FDR Confidence = High and from more than one unique peptide. For the downstream analysis, a pseudocount of 0.1 was added to all the normalized protein abundances. The differential global abundance changes were determined using the *t*-test. A *t*-test was performed to determine the upregulated (FC > 1, *p* < 0.05) or downregulated proteins (FC < 1, *p* < 0.05). The PISA abundance values were normalized to the global abundance values to mitigate the changes in the DZNep-induced protein abundance. A *t*-test was performed to determine the thermally stabilized (FC > 1, *p* < 0.05) or destabilized proteins (FC < 1, *p* < 0.05). To compare the global Kme peptide changes, the Kme peptide abundance was normalized to the protein abundance.

For the bioinformatic and data visualization, the following packages were used: *ggplot2* (volcano plots), org.HS.*eg*.db, *Ggally* (Pearson Correlation), *clusterProfiler* (Gene Ontology), *ggsimplestats* (Chi Square analysis), *ggVennDiagram* (Venn Diagram), *ggpubr* (bargraphs), and *UpsetR* (Upset plots). The GO term GO:0005524 was used to identify the annotated ATP binders, while the GO term GO:0008168 was used to identify the annotated methyltransferases. PhosphositePlus© and the previous data from our lab were used to identify the previously detected lysine methylation sites [1,24]. The STRING database was used to visualize the protein clusters [25]. The GO term analysis was performed using clusterprofiler with the global proteome or transcriptome as the background on differentially abundant proteins or transcripts [26,27].

## 3. Results

### 3.1. Conducting PISA, Global Proteomics, and Transcriptomics on DZNep-Treated Cells

To determine the optimal experimental parameters for DZNep treatment, we monitored the impact of DZNep on global protein lysine methylation. HEK293T cells treated with DZNep across a concentration gradient at 24 and 48 h had a substantial decrease in lysine trimethylation (Kme3) and an increase in dimethylation (Kme2) (Appendix A). This effect was more pronounced at 48 h, and more Kme3 sites were absent upon treatment with 100 μM of DZNep. For all the subsequent experiments, the HEK293T cells were treated with 100 μM DZNep for 48 h.

Inhibiting the production of SAM will impact transcription, which has previously been investigated using DZNep treatment. In addition, the inhibition of SAM production will impact the expression levels of proteins and the thermal stability of proteins. We used the Proteome Integral Solubility Alteration (PISA) assay and RNA sequencing (RNAseq) to investigate the extent to which each is altered upon DZNep treatment (Figure 1). It is important to note that PISA enables the quantification of global protein levels and protein thermal stability in a single experiment. Following treatment, an equal number of cells (1 × 10^6^) from the DZNep- and vehicle-treated samples (n = 4) were collected for RNAseq. The remaining cells were aliquoted into nine equal fractions; eight were heated across an eight-temperature gradient (39.4–61.4 °C), with the ninth remaining tube designated for global proteomics. Following three snap-freeze cycles, the soluble protein was collected from each sample. The PISA sample was created by pooling an equal volume of soluble protein from each heat-treated sample. We verified that all the DZNep experiments perturbed the lysine methylome similarly (Appendix A). This resulted in four DZNep global samples, four DZNep PISA samples, four control global samples, and four control PISA samples combined to create a single 16-plex sample for analysis by LC-MS/MS.

The initial analysis revealed consistent measurements of the global gene and protein expression among all four biological replicates. The RNAseq biological replicates displayed a high correlation (R > 0.9, *p*-value < 0.05, and *n* = 19,051) and clustered together by PCA plot (Appendix A). In the global proteomics analysis, 4890 proteins were detected. This includes the detection of 57,159 peptides from 267,975 peptide spectrum matches (PSMs). After filtering for proteins with a high FDR (*p* < 0.05) and more than one unique peptide, 3585 proteins were quantified. The Pearson correlation between the biological replicates for protein level quantification was greater than 0.99 (Appendix A), showing a high degree of reproducibility. In addition, the replicates clustered together by PCA plot (Appendix A). Altogether, these results demonstrate that the biological replicates are similar.

### 3.2. Global Proteomics and Transcriptomics Reveal Changes in Metabolic Processes and Nucleosome Organization

We then assessed the impact of DZNep on global gene and protein expression. At the transcript level, 2611 transcripts were downregulated while 2092 were upregulated in DZNep-treated cells compared to the vehicle-treated cells (Figure 2A). Upon DZNep treatment, there was a decrease in the transcripts related to metabolic processes, including monocarboxylic acid metabolic processes (Appendix A). In addition, there was a decrease in the transcripts related to movement, particularly cilium mobility. The transcripts upregulated upon DZNep treatment were enriched for DNA binding and nucleosome organization proteins. At the protein level, 458 proteins decreased while 534 proteins increased in abundance (2593 were unchanged) in the DZNep-treated cells compared to the vehicle-treated control (Figure 2B). The gene ontology (GO) analysis revealed that the proteins with increased abundance were enriched for processes, including RNA processing, DNA binding, and chromatin processing. In contrast, the proteins with decreased abundance were enriched for metabolic processes, such as oxoacid, carboxylic acid, and amino acid metabolic processes (Appendix A). There was also an enrichment for the transcripts that encode the proteins involved in signaling pathways and adhesion. There was an overall low but significant correlation between the protein and transcript levels (R = 0.52 and *n* = 3551) (Figure 2D). Interestingly, only 10.7% of the proteins that had an increase in protein abundance also had an increase in transcript levels. Similarly, 25% of the proteins with decreased abundance also had a decrease in transcript levels (Figure 2E and Appendix A). This indicates that the global proteomic changes observed are not solely due to direct changes in transcription. Overall, these data reveal a broad rewiring of global transcription and protein expression upon DZNep treatment.

### 3.3. DZNep Impacts the Thermal Stability of Hundreds of Proteins

We then investigated the thermal stability changes upon DZNep treatment. To account for protein abundance differences between the samples, the PISA abundance values were normalized to the global proteome abundance values, and the fold change between the control and DZNep normalized PISA values was calculated. The DZNep treatment resulted in the thermal stabilization of 91 proteins and destabilization of 44 proteins (Figure 2C). A comparison of the overlap between the thermal stability, protein abundance, and transcript abundance shows that 70% (94 out of 135) of the proteins with altered thermal stability upon DZNep treatment also had a significant abundance change at the transcript or protein level (Figure 2E). A total of 49% (66 out of 135) of the thermal stability changes are solely explained by changes in the protein abundance. Interestingly, this includes 24 proteins that were thermally stabilized but had increased protein abundance (Figure 2E). Overall, these data show that there are both unique and overlapping impacts on the transcript, protein, and protein thermal stability in DZNep-treated cells. We next turned to identify how the added dimension of protein thermal stability can help reveal the potential targets of DZNep.

### 3.4. DZNep Globally Impacts Components of the Methionine Cycle

DZNep is reported to inhibit AHCY via competitive inhibition; nonetheless, some studies suggest it binds to many other proteins besides AHCY [15]. Interestingly, we did not detect significant changes in the transcript, protein abundance, or thermal stability for AHCY upon DZNep treatment. While it is common that the small-molecule engagement of a protein can impact thermal stability, it is possible DZNep binding to AHCY does not impact its thermal stability. However, the protein abundance of several methionine cycle components (MTR, MATIIA, and MATIIB) were significantly lower following DZNep treatment (Figure 3A,B). While the MTR was also decreased at the transcript level, the MATIIA and IIB were not, suggesting that the decrease may be regulated at the protein level. The MTR, which converts homocysteine to methionine using 5′-methyltetrahydrofolate, was also thermally stabilized upon treatment with DZNep (Figure 3A,B). Overall, while we do not see stability or abundance changes to the reported DZNep target, components of the target pathway are significantly impacted at the transcript, protein, and thermal stability levels.

### 3.5. DZNep Impacts the Thermal Stability of the Aminoacyl-tRNA Synthetase Multienzyme Complex

The thermally stabilized proteins had an enrichment for members of the aminoacyl-tRNA synthetase multienzyme complex (MSC). A total of 5 of the 11 complex members were stabilized (LARS1, IARS1, EPRS1, MARS1, and AIMP2) (Figure 4A,B). LARS1, IARS1, MARS1, and AIMP2 also exhibited a decrease in protein abundance, and AIMP2 had an additional decrease in transcript levels. The MSC assembles into a V-shaped complex. AIMP2, EPRS1, IARS1, and LARS1 are found at the base of the V-shaped complex, while MARS1, DARS1, and QAR1S assemble into one arm, and KARS1 and RARS1 comprise the other arm [28]. Interestingly, all the components in the base are thermally stabilized. In addition, AIMP2 is responsible for mediating protein–protein interactions within the complex and is indispensable for its formation [28,29,30].

All the components of the MSC can bind to ATP. Because DZNep is an adenosine analog, their stabilization may be due to the off-target binding of DZNep. Additionally, MARS1, LARS1, and IARS1 all have multiple observed methylation sites [1,24], implying that a perturbation to the SAM/SAH ratio may impact the methylation of the components of the MSC, leading to the observed global decrease and thermal stabilization. A previous study showed that the loss of AIMP2 and individual tRNA synthetases does not affect global protein synthesis [31], suggesting that the loss of AIMP2 induced by DZNep treatment may not explain the changes in protein levels observed in our study. However, it is not understood how the stability and protein level changes observed for multiple components of the MSC complex in our experiments might impact protein synthesis.

### 3.6. DZNep Impacts the Thermal Stability of ATP Binders and Methyltransferases

Because DZNep interacts with adenosine binders [15], we looked at the potential off-target effects of DZNep interactions with ATP binders and methyltransferases. DZNep significantly impacted the global abundance of Ensembl annotated ATP binders (*n* = 467, Chi square *p* < 0.05) (Appendix A). Specifically, it led to the decrease in the global protein abundance of ATP binders. There was also a trend toward the stabilization of the ATP binders, though not significant (Chi square, *p* = 0.07) (Appendix A). In contrast, DZNep did not impact the global abundance of the methyltransferases but did significantly impact their thermal stability (*n* = 66, Chi square, *p* < 0.05) (Appendix A). Interestingly, there was a group of proteins that only exhibited thermal stability changes. These include four methyltransferases: LCMT1 (Leucine carboxyl methyltransferase 1), DNMT1 (DNA methyltransferase 1), ASMTL (probably bifunctional dTTP/UTP pyrophosphatase/methyltransferase protein), and NTMT1 (N-terminal Xaa-Pro-Lys N-methyltransferase 1) (Figure 5A). It was previously shown that DNMT1 protein abundance is not impacted by DZNep, which we recapitulate here; however, PISA was able to detect that DZNep thermally destabilizes DNMT1 [14]. One possible mechanism could be the impact of the SAM/SAH ratio on the activity of DNMT1. Additionally, DNMT1 was previously reported to be methylated at several lysine residues, and DZNep treatment may alter the levels of DNMT1 lysine methylation and impact DNMT1 thermal stability [32].

### 3.7. DZNep Treatment Alters the Abundance of Methylated Proteins and Peptides

The detection of methylated lysine via MS-based proteomics has historically relied upon antibody enrichment. An analysis of the changes in lysine methylation on whole-cell lysates using these antibodies showed drastic changes in the lysine methylation (Appendix A). We investigated if we could detect and quantify changes in the global abundance of peptides with methylated lysine residues by MS directly in the PISA experiment. A total of 610 unique methylated peptides from 388 unique proteins were detected, and 278 unique Kme sites were quantified from 247 unique proteins. Over half of the detected and quantified Kme sites were trimethylated, while less than 20% were monomethylated. The abundance of peptides with lysine methylation was normalized to the global abundance of the corresponding protein to account for the changes in the global protein abundance. In total, 16 Kme sites had an increase in peptide abundance with DZNep treatment, while 24 sites had a decrease in peptide abundance (Figure 5B).

The thermal stability of the methylated peptides was then assessed. To calculate the fold change between the normalized PISA values of the Kme sites, the peptide PISA abundance was normalized to the global abundance of that peptide. Only two Kme sites were thermally stabilized (CDK6 K3me3 and TAC3 K9me3) and three sites were thermally destabilized (PGK1 K139me3, VCP K315me3, and PGK1 K141 me3) (Figure 5C). Interestingly, only TACC3 had an increased protein abundance and only CDK6 was stabilized.

To assess the impact of DZNep on the thermal stability of methylated proteins, we examined the impact of DZNep treatment on proteins with reported lysine methylation sites (PhosphoSite Plus and previous studies) [1,24]. In total, 1631 proteins had previously been observed to be methylated and exhibited a trend toward being more likely to be differentially abundant compared to non-methylated proteins (Chi-square, *p* = 0.058) (Appendix A). Proteins reported to be methylated were also more likely to have altered thermal stability than non-methylated proteins upon DZNep treatment (Chi-square, *p* < 0.05). In addition, proteins were more likely to be destabilized upon DZNep treatment if there was a reported Kme site (Appendix A). The same number of randomly selected proteins did not exhibit this trend, demonstrating that DZNep treatment did impact the thermal stability of methylated proteins (Appendix A). Interestingly, 57% of the methylated, destabilized proteins have the capacity to bind RNA (GO term analysis, *p* < 0.05). Nearly all studies that have profiled lysine methylation using MS-based approaches have noted an enrichment of RNA processing proteins, which is consistent with these data [33,34,35].

## 4. Discussion

In this work, we uncover the impact of DZNep treatment in HEK293T cells on the transcriptome, proteome, and proteome thermal stability. We show that DZNep treatment leads to broad transcriptomic and proteomic changes, though less than a quarter of the proteomic changes can be explained by transcriptomic changes. Using PISA, we demonstrate that over 100 proteins undergo thermal stability changes following DZNep treatment, and 41 of these are independent of any alteration in the corresponding transcript or protein level.

DZNep has frequently been studied as a potential therapeutic compound. DZNep treatment in cancer cell lines leads to a decrease in cellular proliferation [12,13]. Additionally, DZNep has been used as an antiviral compound, with the rationale that inhibiting AHCY would impact the 5′ and 2′-O-ribose methylation cap of viral RNA by decreasing the cellular levels of SAM [8,36]. Considered a broad methyltransferase inhibitor, we detected thermal stability changes for four methyltransferases: DNMT1, LCMT1, NTMT1, and ASMT1. Given the low number of methyltransferases detected (*n* = 60), it is possible that DZNep did alter the thermal stability of additional methyltransferases below the detection limit in the current experiment.

We observed decreased abundance in components of the methionine cycle, supporting the notion that a major mechanism of action of DZNep is due to the disruption of the methionine cycle. Interestingly, the putative direct target of DZNep, AHCY, did not undergo a change in protein abundance or stability. However, DZNep treatment also resulted in the global reduction and thermal stabilization of several key subunits of the aminoacyl tRNA synthase multienzyme complex, including AIMP2, without which the complex dissembles. The abundance and thermal perturbation of both the methionine cycle and the MSC complex highlight the cumulative impact a single compound can have on cellular functions by impacting multiple pathways.

Recently, studies have reported the impact of PTMs (phosphorylation and O-linked N-acetylglucosamine) on protein thermal stability [16,17]. A critical component in the case of phosphorylation was the accurate and robust detection of the modified and unmodified peptides. For the O-linked N-acetylglucosamine study, a clear system to globally alter the modification allowed for clear conclusions to be drawn about the impact of the modification on protein thermal stability. Lysine methylation is a widespread PTM, but annotating specific sites with specific functions has not kept pace with the expansion of the number of sites identified. The approach taken in this work helps demonstrate the potential and technical challenges of using thermal proteome stability assays to curate putative functional lysine methylation sites. Notably, we identified thermal stability changes in several proteins known to be methylated under conditions where global lysine methylation is altered. In future studies, applying this methodology to other cell types and conditions will be useful in dissecting functional lysine methylation signaling pathways [3]. It will be particularly interesting in cancer cells, which often have dysregulation of the writers and erasers of lysine methylation in which non-histone lysine methylation has often not been fully investigated. However, detecting and quantifying lysine methylation sites is still difficult in untargeted proteomics experiments. Furthermore, DZNep impacts proteins and pathways unrelated to lysine methylation, and there is not a clear strategy equivalent to removing the entire pathway as shown for O-GlcNAc. Nonetheless, the use of PISA and other thermal proteome profiling approaches offers promise for studying the impact of post-translational modifications.

## Figures and Tables

**Figure 1 biomolecules-14-00817-f001:**
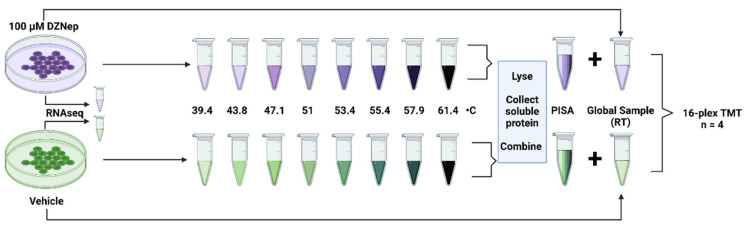
A workflow overview of the combined transcriptome, proteome, and thermal stability analysis. The HEK293T cells were treated with either 100 µM DZNep or vehicle control for 48 h. A total of 1 million cells were collected for the RNAseq analysis. An equal number of cells were then aliquoted into 10 microtubes. Eight tubes were subjected to heat treatment across a temperature gradient as the PISA samples. All the aliquots were then lysed via 3× freeze–thaw cycles. An equal volume of the heat-treated soluble protein was combined into a single microtube per sample. The soluble protein of the non-heat-treated protein was collected as the global proteomics sample. In total, there were four control global proteome samples, four control PISA samples, four DZNep global proteome samples, and four DZNep PISA samples. These were digested using trypsin/LysC, multiplexed using Tandem Mass Tag (TMT) labels, and analyzed using LC-MS/MS. Created with BioRender.com.

**Figure 2 biomolecules-14-00817-f002:**
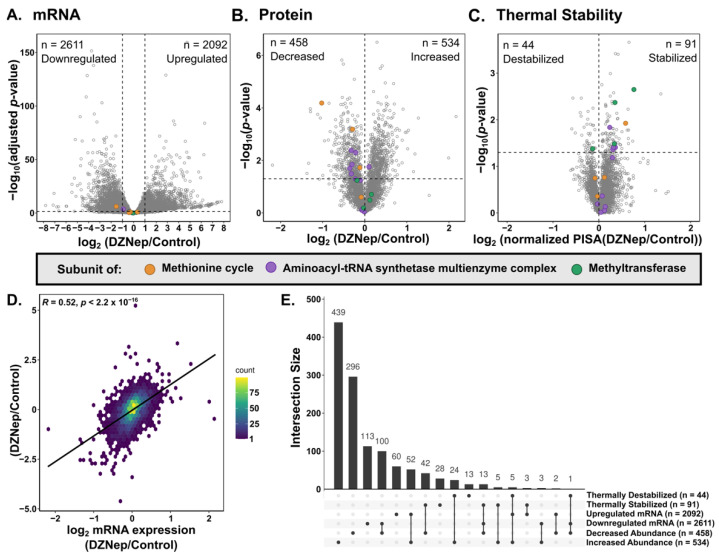
Volcano plots of the fold change vs. −log10 of the (**A**) adjusted *p*-value for mRNA and (**B**,**C**)−log10 of the *p*-value for the protein and thermal stability upon DZNep treatment. The horizontal dashed lines indicate the cutoff of an adjusted *p*-value ≥ 0.05. For the mRNA, the vertical dashed lines indicate a fold change cutoff ≥ |2|. The thermal stability is shown after normalizing the PISA abundance values to global protein abundance. (**D**) The density correlation plot between the observed transcript and protein abundances (Pearson correlation; *n* = 3551). The x-axis is the log_2_ of the fold change between the DZNep and control normalized mRNA levels. The y-axis is the log_2_ fold change between the DZNep and control protein abundances. (**E**) The upset plot visualizing the overlap between the changes in the mRNA, protein, and thermal stability.

**Figure 3 biomolecules-14-00817-f003:**
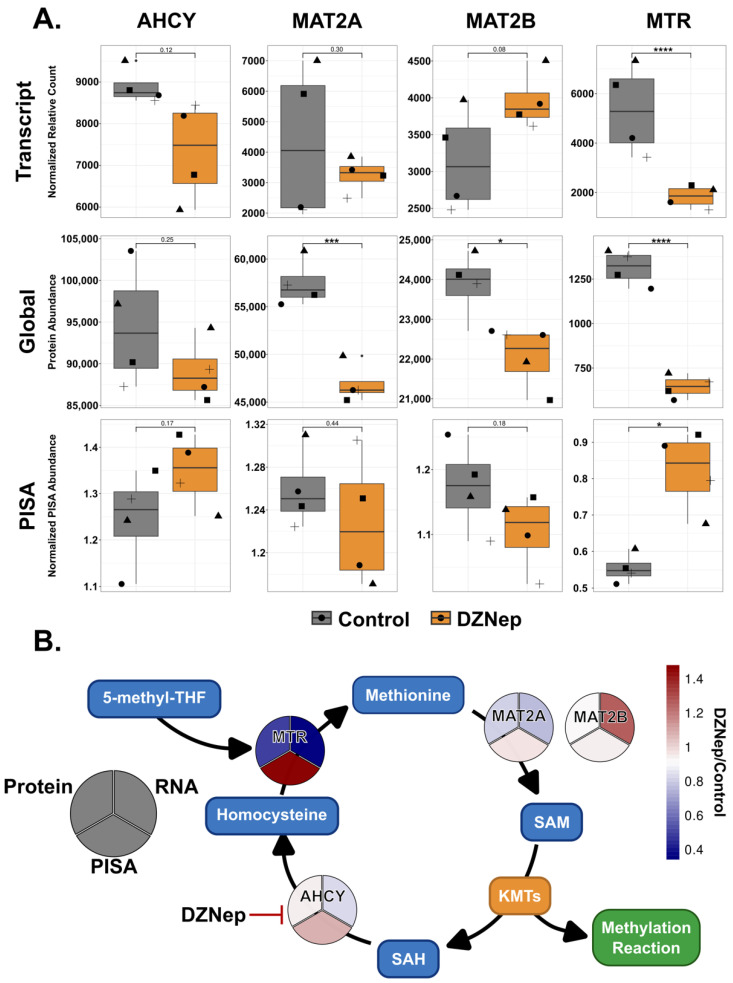
DZNep treatment affects components of the methionine cycle. (**A**) Bar graphs visualizing the difference in the transcript, protein, and normalized PISA abundances between the control (grey bars) and DZNep-treated samples (orange bars) for the genes as indicated. Shapes represent the different replicates (*n* = 4; *t*-test, * = *p* < 0.05, *** = *p* < 0.001, and **** = *p* < 0.0001). (**B**) A schematic of the methionine cycle. The circle heatmaps represent the fold change between the DZNep- and control-treated samples within the indicated dataset (RNA, protein, and thermal stability). The blue boxes represent products and cofactors.

**Figure 4 biomolecules-14-00817-f004:**
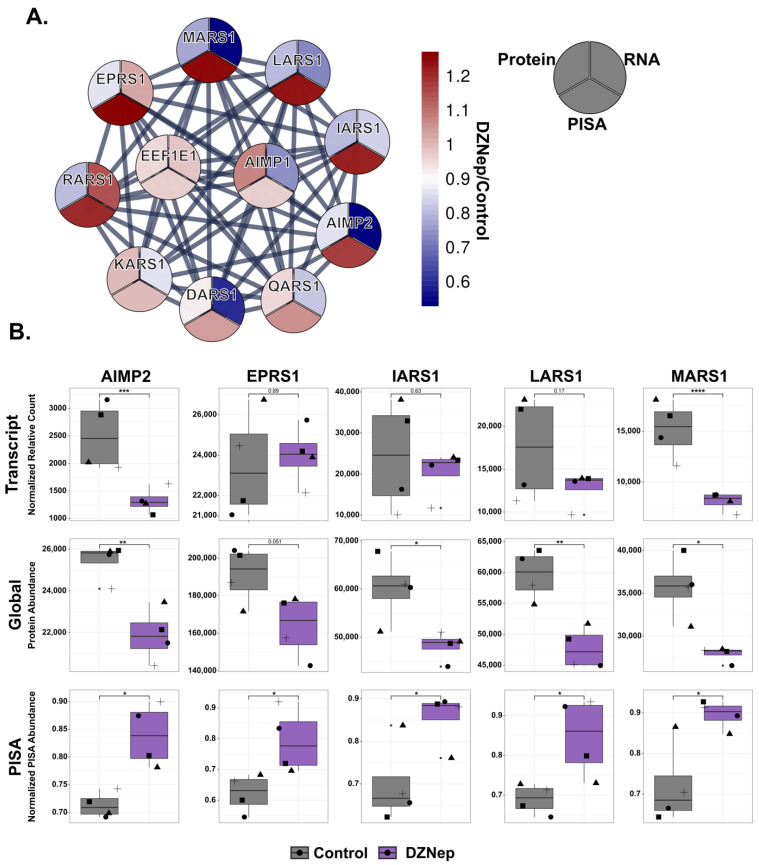
DZNep treatment stabilizes the components of the aminoacyl-tRNA synthetase multienzyme complex (MSC). (**A**) The STRING network of the subunits of the MSC. The circle heatmaps visualize the fold change (DZNep/control) in the different MSC subunits from each dataset (RNA, protein, and thermal stability) (**B**) Bar graphs visualizing the difference in transcript, protein, or PISA abundances between the control (grey bars) and DZNep-treated samples (purple bars) for the genes as indicated. Shapes represent different replicates(*n* = 4; *t*-test, * = *p* < 0.05, ** = *p* < 0.01, *** = *p* < 0.001, and **** = *p* < 0.0001).

**Figure 5 biomolecules-14-00817-f005:**
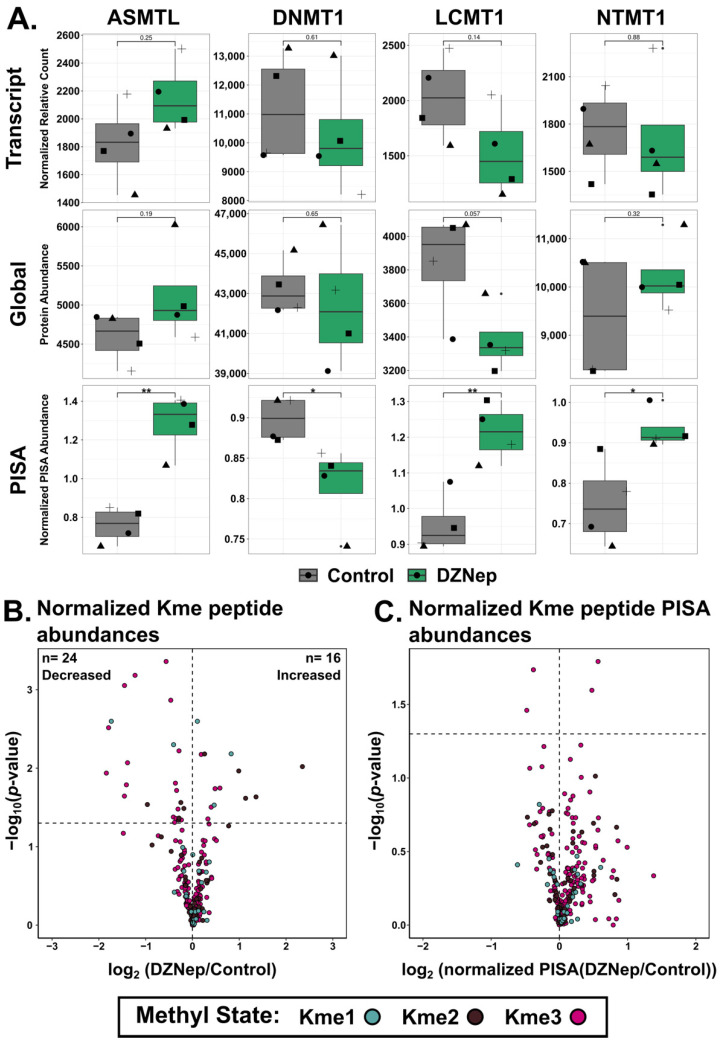
(**A**) Bar graphs visualizing the difference in the transcript, protein, or PISA abundances between the control (grey bars) and DZNep-treated samples (green bars) for the indicated genes. Shapes represent different replicates (*n* = 4; *t*-test, * = *p* < 0.05, ** = *p* < 0.01). (**B**) Volcano plots of the fold change vs. −log10 of the *p*-value for lysine-methylated peptides and (**C**) the thermal stability of lysine-methylated peptides. Peptides carrying monomethyl, dimethyl, or trimethyl lysine are indicated in green, brown, and magenta, respectively.

## Data Availability

The code used to analyze these data is accessible at https://github.com/caberryhill/DZNep-PISA. The RNAseq data, including the raw sequencing data and normalized read counts, are available through the National Center for Biotechnology Information Gene Expression Omnibus using accession GSE268629. All the raw mass spectrometry data and processed Proteome Discoverer result files are uploaded to the MassIVE repository with accession MSV000094902.

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
