# Peer review of "Protein Thermal Stability Changes Induced by the Global Methylation Inhibitor 3-Deazaneplanocin A (DZNep)"

_biomolecules, 2024, doi:10.3390/biom14070817_

Round 1

Reviewer 1 Report

Comments and Suggestions for Authors

The manuscript "Protein Thermal Stability Changes Induced by the Global Methylation Inhibitor 3-deazaneplanocin A (DZNep)" by Berryhill et al., describes the impact of DZNep treatment on the transcriptome, proteome, and thermal stability of proteins in HEK293T cells. The manuscript is written very well. The investigation strategy is well designed and pursued, and the results were described appropriately. However, I have a few minor comments as below:

  1. In the Abstract, line 15, “and reducing the level of…SAM.” Is it appropriate to say that DZNep reduces SAM level? For SAM, S-adenosyl-L-methionine should be used rather than S-adenosylmethionine.
  2. In the Materials and Methods,

a. Line 94-95, “Cells were….. described below.” doesn’t make proper sense.

b. Line 99, better to write Lysis buffer (composition) together.

c. Line 111, typo ‘DNZep’.

d. Line 115, the temperature gradient for heat treatment were set for 8 points. These set points don’t follow any regular increment of temperature. How did you decide these temperature values to be measured in PISA?

  1. Future directions of the study have not been mentioned. The study was performed on HEK293T cells. Have you (or others) done such study on cancer cell lines? What difference would you expect?

Author Response

Comment 1:In the Abstract, line 15, “and reducing the level of…SAM.” Is it appropriate to say that DZNep reduces SAM level? For SAM, S-adenosyl-L-methionine should be used rather than S-adenosylmethionine

Response 1: We thank the reviewer for this comment. We agree that “reduces the level of SAM” is likely an indirect impact of DZNep treatment. To avoid emphasizing this conclusion, we have removed this comment. We have corrected S-adenosylmethionine to S-Adenosyl-L-methionine throughout the entire manuscript.

Comment 2: In the Materials and Methods,

  1. Line 94-95, “Cells were….. described below.” doesn’t make proper sense.
  2. Line 99, better to write Lysis buffer (composition) together.
  3. Line 111, typo ‘DNZep’.
  4. Line 115, the temperature gradient for heat treatment were set for 8 points. These set points don’t follow any regular increment of temperature. How did you decide these temperature values to be measured in PISA?

Response 2: Thank you for these helpful suggestions. The following revisions have been made:

 a. Line 95-96: Corrected to “Cells were collected 48 hours later and immediately subjected to the Proteome Integral Solubility Assay.”

 b. Line 99: Corrected to: “Cells were grown until 90% confluent, pelleted, and protein was extracted by resuspending in lysis buffer (10 mM PIPES pH 7, 300 mM sucrose, 100 mM NaCl, 3 mM MgCl2, 0.1% Triton X-100, protease inhibitors (Thermo Scientific - Pierce #A32965), and nuclease (Thermo Scientific – Pierce #88700)) and incubating for 15 minutes on ice.

 c. Line 112: Corrected to “DZNep”

 d. Line 115: These temperatures were chosen to encompass the melting curve of the human proteome. Temperatures were selected from the BioRad thermocycler temperature gradient, with more temperature points selected within a typical melting curve.

Comment 3: Future directions of the study have not been mentioned. The study was performed on HEK293T cells. Have you (or others) done such study on cancer cell lines? What difference would you expect?

Response 3: Thank you for pointing this out. This type of approach has not been used in cancer cells, but is a clear future direction. We have added the following sentences to the discussion to make this point:

Line 499-503

“In future studies, applying this methodology to other cell types and conditions will be useful in dissecting functional lysine methylation signaling pathways. It will be particularly interesting in cancer cells, which often have dysregulation of the writers and erasers of lysine methylation in which non-histone lysine methylation has often not been fully investigated [3].”

Reviewer 2 Report

Comments and Suggestions for Authors

In this study, the authors investigated the impact of DZNep on the transcriptome, proteome, and thermal stability of proteins. The paper is well-written and organized. I think it's suitable to be published in this Journal. However, I have some confusions about the materials used in this work. 1) How do you evaluate the stability of protein materials? 2) Do you have some chemical information analysis on protein stability. For example, the change in secondary structure of protein? 3) Have you tried to visualize the protein materials? For instance, the actual morphology?

Author Response

Comment 1:

In this study, the authors investigated the impact of DZNep on the transcriptome, proteome, and thermal stability of proteins. The paper is well-written and organized. I think it's suitable to be published in this Journal. However, I have some confusions about the materials used in this work. 1) How do you evaluate the stability of protein materials? 2) Do you have some chemical information analysis on protein stability. For example, the change in secondary structure of protein? 3) Have you tried to visualize the protein materials? For instance, the actual morphology?

Response 1:

The PISA approach used in this study measures changes in protein abundance as a result of changes in protein thermal stability. After applying heat to samples, the soluble proteins are quantified by mass spectrometry, and proteins are determined to be stabilized or destabilized from a resulting increase or decrease in abundance compared to the control. Previous work in the field has shown that multiple factors can impact a protein's melting temperature or thermal stability including direct binding of a compound, post translational modifications, folding, and protein-protein interaction changes (see ref 16-19, 21). In the present study, we do not attempt to visualize or use chemical information to analyze the protein stability. Such analysis would be interesting for proteins with altered thermal stability but is beyond the scope of our current work.